# Behavioral insights during the COVID-19 pandemic in the Federation of Bosnia and Herzegovina: the role of trust, health literacy, risk and fairness perceptions in compliance with public health and social measures

Šeila Cilović-Lagarija[1], Sarah Eitze[2,3]*, Siniša Skočibušić[1,4], Sanjin Musa[1], Stela Stojisavljević[5], Haris Šabanović[6], Faris Dizdar[7], Mirza Palo[7], Dorit Nitzan[8], Miguel Telo de Arriaga[9,10], Martha Scherzer[11], Benjamin Curtis[11], Katrine Bach Habersaat[11]

1 Institute for Public Health FB&H, Sarajevo, Bosnia and Herzegovina, 2 Institute for Planetary Health Behavior, University of Erfurt, Germany, 3 Health Communication, Implementation Research, Bernhard Nocht Institute for Tropical Medicine, University of Hamburg, Hamburg, Germany, 4 Faculty of Medicine, University of Mostar, Mostar, Bosnia and Herzegovina, 5 Faculty of Medicine, University of Banja Luka, Banja Luka, Bosnia and Herzegovina, 6 Valicon, Sarajevo, Bosnia and Herzegovina, 7 Country Office in Bosnia and Herzegovina, World Health Organization, Sarajevo, Bosnia and Herzegovina, 8 School of Public Health & Chair, Food Systems, One Health and Resilience, Ben Gurion University of the Negev, Israel, 9 Direção- Geral da Saúde, Portugal Instituto de Saúde Ambiental, Faculdade de Medicina, Universidade de Lisboa, Lisboa, Portugal, 10 CRC-W—Católica Research Centre for Psychological, Family and Social Wellbeing, Universidade Católica Portuguesa, Lisbon, Portugal, 11 World Health Organization Regional Office for Europe, Copenhagen, Denmark

* sarah.eitze@uni-erfurt.de

## Abstract

### Background and aim

Public health and social measures (PHSM) are critical aspects of limiting the spread of infections in pandemics. Compliance with PHSM depends on a wide range of factors, including behavioral determinants such as emotional response, trust in institutions or risk perceptions. This study examines self-reported compliance with PHSM during the COVID-19 pandemic in the Federation of Bosnia and Herzegovina (FBIH).

### Materials and methods

We analyze the association between compliance and behavioral determinants, using data from five cross-sectional surveys that were conducted between June 2020 and August 2021 in FBIH. Quota-based sampling ensured that the 1000 people per wave were population representative regarding age, sex, and education level based on the data from the latest census in Bosnia and Herzegovina. One-way analysis of variance (ANOVA) was used to identify significant changes between studies on determinants and PHSM measures. Regression was used to find relations between behavioral determinants and PHSM.

**Data availability statement:** In the following, anonymized OSF folder, the rwa data and the complete analysis script is provided, following open data and open analysis standards for easy replication. https://osf.io/5vr37/?view_only=c3ec4c6132f04c709a6cde333c3fd31b

**Funding:** The author(s) received no specific funding for this work.

**Competing interests:** The authors have declared that no competing interests exist.

## Results

Participants reported strong emotional responses to the rapid spread of the virus and its proximity to them. Risk perception was spiking in December 2020 when rates of infection and death were particularly high. Trends in policy acceptance were divergent; participants did not rate PHSM as exaggerated, but perceived fairness was low. Trust in institutions was low across all waves and declined for specific institutions such as the health ministry. In five wave-specific regression analyses, emotional response ($\beta_{min/max} = .11^*/.21^*$), risk perception ($\beta_{min/max} = .06/.18^*$), policy acceptance ($\beta_{min/max} = .09/.20^*$), and trust in institutions ($\beta_{min/max} = .06/.21^*$) emerged as significant predictors of PHSM.

## Conclusions

This study contributes to the body of research on factors influencing compliance with PHSM. It emphasizes the importance of behavioral monitoring through repeated surveys to understand and improve compliance. The study also affirms the impact of public trust on compliance, the risk of eroding compliance over time, and the need for health literacy support to help reinforce protective behaviors.

## Introduction

Evidence-based public health and social measures (PHSM) were introduced around the world during the COVID-19 pandemic [1,2]. The PHSM measures included washing hands regularly, wearing masks, quarantine and isolation in case of certain conditions, school closures, limiting public events, using tissues when sneezing, cleaning and disinfecting objects, and social distancing. The World Health Organization recommended that governments implement a particular set of PHSM, considering epidemiological data, health system capacity, other data sources, and population-level risk perception and behaviors [3]. Risk perception and behaviors are essential for understanding compliance with PHSM, but in turn, they are influenced by several other factors, including individuals' perceptions of the consistency, competence, fairness, and objectivity of governmental authorities [4–6]. This study aimed to assess the behavioral determinants that influence compliance with PHSM in the Federation of Bosnia and Herzegovina (FBIH) during the first year of the COVID-19 pandemic 2020-2021.

Research suggests that when PHSM are communicated in an easily understood way, through trusted and accessible channels, it aids people in making informed choices to comply and protect themselves and their communities [7,8]. To provide a tool for countries to understand these population-level perceptions and behaviors and how they might influence PHSM compliance, the WHO Europe Regional Office worked together with a team at the University of Erfurt to develop a COVID-19 behavioral insights survey tool. This tool included a standard template protocol and questionnaire for adaptation by public health authorities at country or local levels.

Both PHSM policies, such as cancellation of mass gatherings and complementary behavioral research monitorings were implemented in the Federation of Bosnia and Herzegovina (FBIH), tailored and adapted to the specific epidemiological situation that changed over the course of the pandemic ([9–11]. The survey was delivered in five discrete waves between the spring of 2020 and the summer of 2021. The aims of the survey were to 1) monitor the behavioral determinants critical for population compliance with the PHSM, including health literacy, risk perceptions, trust, and perception of fairness of the measures; 2) assess population

behavioral changes regarding uptake of PHSM over time; 3) see how those changes relate to behavioral determinants, and contribute to findings on factors encouraging PHSM uptake more broadly; and 4) identify socio-demographic factors (such as sex and age) that help to describe target groups with low compliance and low trust.

This study scrutinizes the results of the behavioral monitoring system in conjunction with self-reported behaviors. The paper begins by outlining the research methodology including the survey measures. Section II analyses the survey data across the five waves. Section III discusses the key behavioral determinants of PHSM uptake in FBIH. Section IV concludes and offers recommendations.

## Materials and methods

### Sample and procedure

The research team fielded five cross-sectional survey waves in the FBIH during different periods of the COVID-19 pandemic in the country between June 2020 and August 2021, reaching a total of N = 5,195 participants, approximately 1,000 unique respondents per wave [12].

### Time of data collection

- first wave 05/06/2020 - 08/06/2020;
- second wave 03/07/2020 - 06/07/2020;
- third wave 19/09/2020 - 22/09/2020;
- fourth wave 04/12/2020 - 07/12/2020; and
- fifth wave 05/08/2021 - 08/08/2021.

Information about the samples and data collections can be seen in Table 1. These surveys enabled us to monitor changes, trends, and patterns in the population's responses and attitudes toward the COVID-19 pandemic. The research company collecting the data distributed the online questionnaires to various population segments, enabling collection of a diverse range of respondents over time. While distribution methods were constant, response rates increased in the later waves (Wave 1: 21%, 4.818 invitations; Wave 4: 54%, 1.958 invitations; Wave 5: 47%, 2.158 invitations). All participants provided written informed consent documented by the data collection company, and the study was approved by the Ethics Committee in the FBIH and the WHO Ethical Review Committee. Participants could terminate their participation at any time. On average, people took around 17 minutes to complete the survey. Participants were allowed to end the questionnaire at any time. However, over all five waves only 8% of participants who started the questionnaires dropped out before giving all answers. Data were analyzed using R Studio. All scripts and analysis outputs are visible on OSF (https://osf.io/5vr37/?view_only=c3ec4c6132f04c709a6cde333c3fd31b) [13].

### Measures

The questionnaire was based on the WHO Regional Office for Europe COVID-19 behavioral insights questionnaire [14] and adapted to the FBIH context [13]. It was designed to obtain information on individuals' opinions about their risk perceptions, knowledge, self-efficacy, confidence in institutions, behaviors, rumors, affect, worry, resilience, trust in/use of information sources, and more. We report on the following subset of seven variables as we are interested in changes over time. In section II we report on the statistical analysis of these variables as behavioral determinants of PHSM uptake.

**Table 1. Sample demographics.**

| | | Wave 1 (n = 1054) | Wave 2 (n = 1000) | Wave 3 (n = 1067) | Wave 4 (n = 1068) | Wave 5 (n = 1006) | |
|---|---|---|---|---|---|---|---|
| **Time of data collection** | | 05 -08. June 2020 | 03.-06. July 2020 | 19.- 22. September 2020 | 04.-07. December 2020 | 05-08. August 2021 | |
| **Age** | Median | 42 (30,55) | 42 (31,55) | 40 (30,52) | 45 (32,55) | 44 (32,53) | $F_{(5190,4)} =$ 5.033, $p < .001$ |
| | Range | 18, 74 | 18, 74 | 18, 74 | 18, 74 | 18, 74 | |
| **Sex** | Male | 500 (47%) | 483 (48%) | 524 (49%) | 514 (48%) | 482 (48%) | $\chi^2(4) = .637$, $p = .959$ |
| | Female | 554 (53%) | 517 (52%) | 543 (51%) | 554 (52%) | 524 (52%) | |
| **Education** | Primary/high school | 794 (75%) | 730 (73%) | 794 (74%) | 802 (75%) | 684 (68%) | $\chi^2(4) = 19.135$, $p < .001$ |
| | University degree | 260 (25%) | 270 (27%) | 273 (26%) | 266 (25%) | 322 (32%) | |
| **Working in healthcare** | yes | 43.5% | 8.3% | 10% | 9% | 9.2% | $\chi^2(4) = 83.393$, $p < .001$ |
| | no | 56.5% | 91.7% | 90% | 91% | 90.8% | |
| **Having own children (in their households)** | yes | 54.3% | 52.6% | 45.2% | 39% | 39.9% | $\chi^2(4) = 83.393$, $p < .001$ |
| | no | 45.7% | 47.4% | 54.8% | 61% | 60.1% | |
| **Living environment (urban vs. rural)** | urban | 52.8% | 54.8% | 64.5% | 62.7% | 44.5% | $\chi^2(4) = 109.34$, $p < .001$ |
| | rural | 47.2% | 45.2% | 35.5% | 37.3% | 55.5% | |

Sample characteristics within the five waves of data collection. Chi-Square tests were used to test if the distribution was significantly different between waves.

**Self-efficacy.** Self-efficacy refers to an individual's belief in their capacity to execute behaviors necessary to achieve specific outcomes [15]. It is a critical factor in several theories of health behavior [16,17] and has been shown to influence infection prevention [18,19]. Two items were used to assess self-efficacy. The first item covered the theoretical efficacy (i.e., knowing how to protect oneself) ranging from 1 (*not at all*) to 7 (*very much so*). The second item covers the behavioral self-efficacy (i.e., avoiding an infection in the current situation) ranging from 1 (*extremely difficult*) to 7 (*extremely easy*). Due to a low correlation of the items (r = .32), we only use the behavioral self-efficacy as a behavioral determinant in the regression analyses.

**Risk perceptions.** In line with the literature [20], perceived susceptibility, severity, and likelihood of contracting the novel coronavirus infection form cognitive risk perceptions. High risk perceptions are associated with increased protective behaviors [20]. On a 7-point Likert scale ranging from 1 (*extremely unlikely/not severe/not susceptible*) to 7 (*extreme likely/very severe/very susceptible*), respondents rated their agreement with each statement. A Cronbach's alpha of 0.74 confirmed this scale's satisfactory internal consistency.

**Emotional response.** Emotions, especially fear, can play a significant role in motivating individuals to adopt prevention behaviors. According to Bradley & Lang [21], dominance, valence and arousal are important dimensions of emotional responses. We collected the emotional response to the pandemic situation with eight items. All were collected on semantic differentials (e.g., Perceiving the virus as 1 *spreading slow* vs. 7 *spreading fast*; Perceiving the situation as 1 *not stressful* vs. 7 *extremely stressful*). Two items, perceiving the pandemic as media-hyped (vs. not) and feeling depressed (vs. not) might cover the results of more complex cognitive and affective processes [22] rather than the initial emotional response dimensions. Reliability analyses showed that those items were not sufficiently correlated with the rest of the emotional response scale. To reach a sufficient Cronbach's alpha (α = .71 [.70;.72]), the two items had to be excluded from the score.

**Trust.** Respondents were asked to rate their level of trust in health professionals, the Ministry of Health, the Institute of Public Health, their employer, schools and churches. Respondents rated their agreement from 1 (*strongly disagree*) to 7 (*strongly agree*). A scale

using trust in health professionals, in the Ministry of Health, and in the Institute for Public Health had very good internal consistency, indicated by a Cronbach's alpha of $\alpha = .87$ [.87;.88].

**Health literacy.** An individual's ability to access, understand, and apply health-related information about their health, disease symptoms, treatments, and preventive behaviors critical for engaging in protective health behaviors [23]. Individuals with higher health literacy are more likely to seek accurate information, understand public health guidelines, and take proactive steps, such as applying PHSM, to prevent illness [24]. In waves 2–4, participants were asked about their perceived ability to find the information they needed about symptoms, treatments, and behavioral recommendations for COVID-19. In wave 5, this scale was extended with six items to COVID-19 vaccination literacy. Respondents rated their agreement from 1 (*strongly disagree*) to 7 (*strongly agree*). Cronbach's alpha indicates excellent internal consistency ($\alpha = .88$ [.88;.89]) in all waves.

**Fairness and policy acceptance.** During the COVID-19 pandemic, restrictions such as curfews, limitations of gatherings, isolation, and school closings were implemented to depress the number of infections. In each round, participants indicated via three items whether they thought the decisions made by politicians were fair, if they found them exaggerated, and if they convinced others that the decisions were right. We summarized these perceptions into the fairness and policy acceptance scale with moderate internal consistency ($\alpha = .64$ [.62;.65]).

**Protective behaviors.** Over all five waves of data collection, participants were asked whether they engaged in a total of seven recommended preventive behaviors, including handwashing for at least 20 seconds, wearing a face mask, maintaining physical distance, and disinfecting surfaces. Respondents answered these questions with a binary answer format in wave 1–4. In wave 5, the answer scale changed into a 7-point scale ranging from 1 (*never*) to 7 (*always*). For comparison, we recoded answers into a binary form for no (from 1–4) vs. yes (5–7) answers. The mean values internal consistency was sufficient ($\alpha = .74$ [.73;.75]).

## Statistical analysis

First, descriptive statistics were calculated for all demographic characteristics and outcome variables to provide an overview of the sample.

Second, we aimed to examine whether the average values of behavioral determinants changed between the five survey waves. To do this, we performed six one-way analyses of variance (ANOVA), one for each behavioral determinant (self-efficacy, risk perceptions, emotional response, trust, health literacy, and fairness and policy acceptance). ANOVA is a statistical method used to compare means across multiple groups—in this case, the five survey waves—and determine if differences are statistically significant. Tukey post hoc tests were applied to identify specific pairwise differences between the cross-sectional samples. Additionally, we performed an ANOVA for protective behaviors to analyze whether they varied between the five waves.

Third, multiple linear regressions were used for each wave to determine which behavioral determinants significantly predicted preventive behaviors. Regression analysis is a statistical technique that identifies the relationship between one dependent variable (e.g., preventive behaviors) and multiple independent variables (e.g., behavioral determinants). To further explore whether socio-demographic characteristics influenced behavioral determinants, additional linear regressions were conducted, with results detailed in Appendix 1. All analyses were performed using R-Studio, and the raw datasets are available for replication in an Open Science Repository. (https://osf.io/5vr37/?view_only=c3ec4c6132f-04c709a6cde333c3fd31b) [13].

## Results

Socio-demographic analysis shows that the age group of 30-49 years constituted the highest number of respondents with 44.5% (2,313), followed by the 50-64 age group with 26.5% (1,380), while females were represented by 51.8% (2,692) and males by 48.2% (2,503). Individuals residing in urban settlements accounted for 56.0% (2,911) of the sample.

Table 1 shows the samples' distributions. In comparison with representative census data, the samples were slightly more educated (university degree in the sample: 25%–32%, university degree in the population: 10,1%) [25], and had slightly more living conditions that included children (living with children in the sample: 39%–54.3%, living with children in the population: 25.2%) [26].

### Self-efficacy

Table 2 shows the results of the ANOVA and Tukey post-hoc tests. For behavioral self-efficacy, there is a significant change between waves. The post-hoc tests show that self-efficacy decreased significantly from wave 1 to wave 3 and wave 4. Self-efficacy was initially measured with two items. Fig 1 shows both items over time. We can see that perceived knowledge about protective measures is higher than behavioral self-efficacy in all waves, and it seems to be a fairly robust perception, as perceived knowledge remains relatively consistent whereas behavioral self-efficacy declines.

### Risk perception

Risk perception, particularly concerning COVID-19, underwent significant fluctuations across multiple waves of data collection. Initially, there was a noticeable upward trend in the perception of risk among participants, steadily increasing through the first four waves. However, an intriguing shift occurred in the fifth wave, where there was a decline in overall risk perception compared to the preceding waves. This deviation from the upward trend observed in earlier phases suggests a dynamic and evolving perception of risk among individuals throughout the course of the study. Throughout these waves, the participants' perceptions across all dimensions of risk displayed consistent increases, signifying a broad-based elevation in risk awareness and concern. Additional analysis focused on psychological determinants

**Table 2. Analysis of variance and tukey post-hoc comparisons for self-efficacy measurements in four waves of data collection.**

**ANOVA**

|  | df | Sum Sq | Mean Sq | F | p |
|---|---|---|---|---|---|
| Wave | 3 | 140 | 46.54 | 14.78 | **<.001** |
| Error | 4185 | 13178 | 3.15 |  |  |

Tukey Post-hoc comparisons

|  |  | $M_{diff}$ | LCI | UCI | $P_{adj.}$ |
|---|---|---|---|---|---|
| **W2 – W1** |  | −0.177 | −0.378 | 0.024 | 0.107 |
| **W3 – W1** |  | −0.341 | −0.539 | −0.142 | **<.001** |
| **W4 – W1** |  | −0.486 | −0.684 | −0.288 | **<.001** |
| **W3 – W2** |  | −0.163 | −0.364 | 0.037 | 0.156 |
| **W4 – W2** |  | −0.309 | −0.510 | −0.109 | **<.001** |
| **W4 – W3** |  | −0.146 | −0.343 | 0.051 | 0.228 |

Results for the analysis of variance with survey wave as independent and self-efficacy (knowing how to protect oneself, avoiding an infection) as the dependent variable. Tukey Post-Hoc tests compare the means between the four waves (W1, W2, W3, W4). Adjusted p-values and 95%-confidence intervals (LCI = lower confidence interval bound; UCI = upper confidence interval bound) show significant differences.

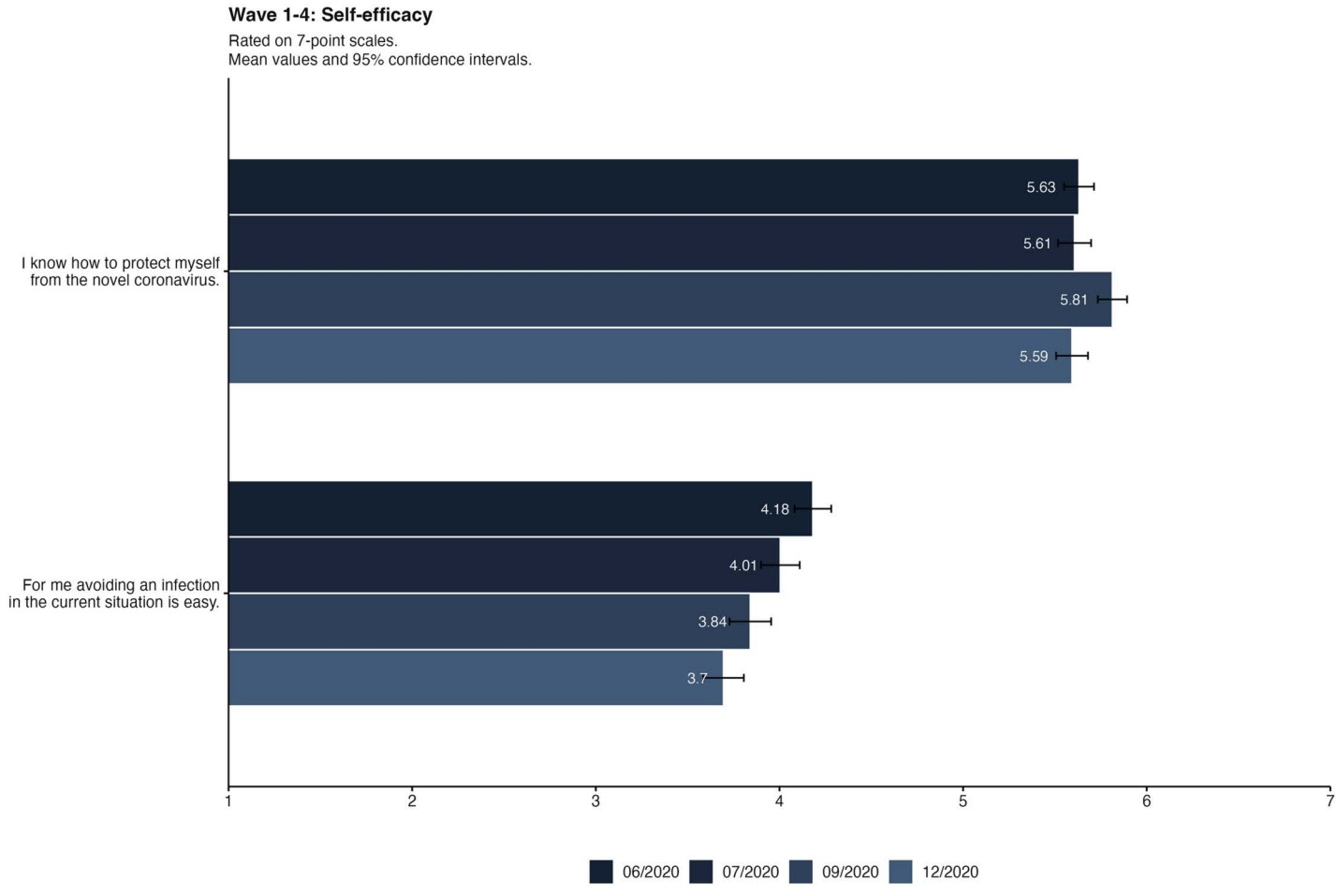

**Fig 1. Self-efficacy items in the four waves of data collection.** Bars displaying mean values and 95%-Confidence Intervals.

**Table 3. Analysis of variance and tukey post-hoc comparisons for risk perceptions in five waves of data collection.**

**ANOVA**

|  | df | Sum Sq | Mean Sq | F | p |
|---|---|---|---|---|---|
| Wave | 4 | 446 | 111.56 | 54.34 | **<.001** |
| Error | 5190 | 10656 | 2.05 |  |  |

Tukey Post-hoc comparisons

|  |  | $M_{diff}$ | LCI | UCI | $p_{adj.}$ |
|---|---|---|---|---|---|
| **W2 – W1** |  | 0.382 | 0.209 | 0.555 | **<.001** |
| **W3 – W1** |  | 0.547 | 0.377 | 0.716 | **<.001** |
| **W4 – W1** |  | 0.901 | 0.732 | 1.071 | **<.001** |
| **W5 – W1** |  | 0.497 | 0.324 | 0.669 | **<.001** |
| **W3 – W2** |  | 0.165 | −0.008 | 0.337 | 0.069 |
| **W4 – W2** |  | 0.519 | 0.347 | 0.691 | **<.001** |
| **W5 – W2** |  | 0.115 | −0.060 | 0.289 | 0.378 |
| **W4 – W3** |  | 0.355 | 0.186 | 0.524 | **<.001** |
| **W5 – W3** |  | −0.050 | −0.222 | 0.122 | 0.933 |
| **W5 – W4** |  | −0.405 | −0.577 | -0.233 | **<.001** |

Results for the analysis of variance with survey wave as independent and risk perception (including susceptibility, severity and probability) as the dependent variable. Tukey Post-Hoc tests compare the means between the five waves (W1, W2, W3, W4, W5). Adjusted p-values and 95%-confidence intervals (LCI = lower confidence interval bound; UCI = upper confidence interval bound) show significant differences.

influencing pandemic behavior reveals several noteworthy associations. Notably, individuals in urban areas or with chronic illnesses tended to exhibit higher levels of risk perception, highlighting how environmental and health-related factors may intensify perceived risk. Conversely, individuals with children tended to demonstrate comparatively lower levels of risk perception. This divergence underscores the potential influence of familial considerations or different risk assessment strategies employed by individuals with children, which warrants further exploration.

The statistical analysis, captured in Table 3, utilizing ANOVA and Tukey post-hoc comparisons, further delineated the nuanced shifts in risk perception across the waves of data collection. The ANOVA results indicated a significant variation in risk perception across the different waves, with subsequent post-hoc comparisons revealing specific pairwise differences between waves, underscoring the dynamic nature of risk perception across the study's duration. In summary, the participants' risk perception related to COVID-19 generally increased over the first four waves before declining in the fifth wave (Fig 2).

### Emotional response

The emotional response to the pandemic was a pivotal aspect examined throughout the five waves of data collection. The scale captured various emotional dimensions, ranging from anxiety to resilience, fear to hope, and stress. Each wave of data collection allowed for a nuanced understanding of how these emotional responses evolved and fluctuated over time, offering insights into the dynamic nature of individuals' reactions to the prolonged impact of the pandemic. In general, the items that the virus was "spreading fast" and "close" produced the strongest emotional reaction among respondents. Most items also displayed a spike in the fourth wave in December 2020.

Table 4 presents an analysis of variance (ANOVA) and Tukey post-hoc comparisons concerning risk perceptions across the different waves of data collection. These statistical analyses elucidate the relationship between emotional responses and risk perceptions, highlighting significant differences and patterns observed over the course of the study.

Fig 3 illustrates the items used to measure emotional responses across all five waves. Notably, items denoted with asterisks within the emotional response scale were integral in comprehending the spectrum and intensity of emotional reactions experienced by participants.

### Trust

Confidence in institutions encompasses trust in doctors, the Ministry of Health, the Institute for Public Health, educational institutions, employers, and religious organizations. Over all five waves there is a tendency of decreased trust, as can be seen from the general model of the analysis of variance in Table 5. However, in the post-hoc analysis using Bonferroni corrections, none of the data collections is significantly different from others because the effect is small. Descriptive differences between all waves and items can be seen in Fig 4. Doctors emerged as the most trusted individuals throughout the pandemic. However, the Ministry of Health and the Institute for Public Health experienced a declining trend in trust during the fall and winter of 2020. Interestingly, churches, while being the least trusted institution in handling the pandemic, exhibited a notable increase in trust over time.

### Health literacy

Data on health literacy was collected in wave 2 to wave 5. Even though the scale changed in the last wave, we conducted an analysis of variance to see if there were any significant changes over

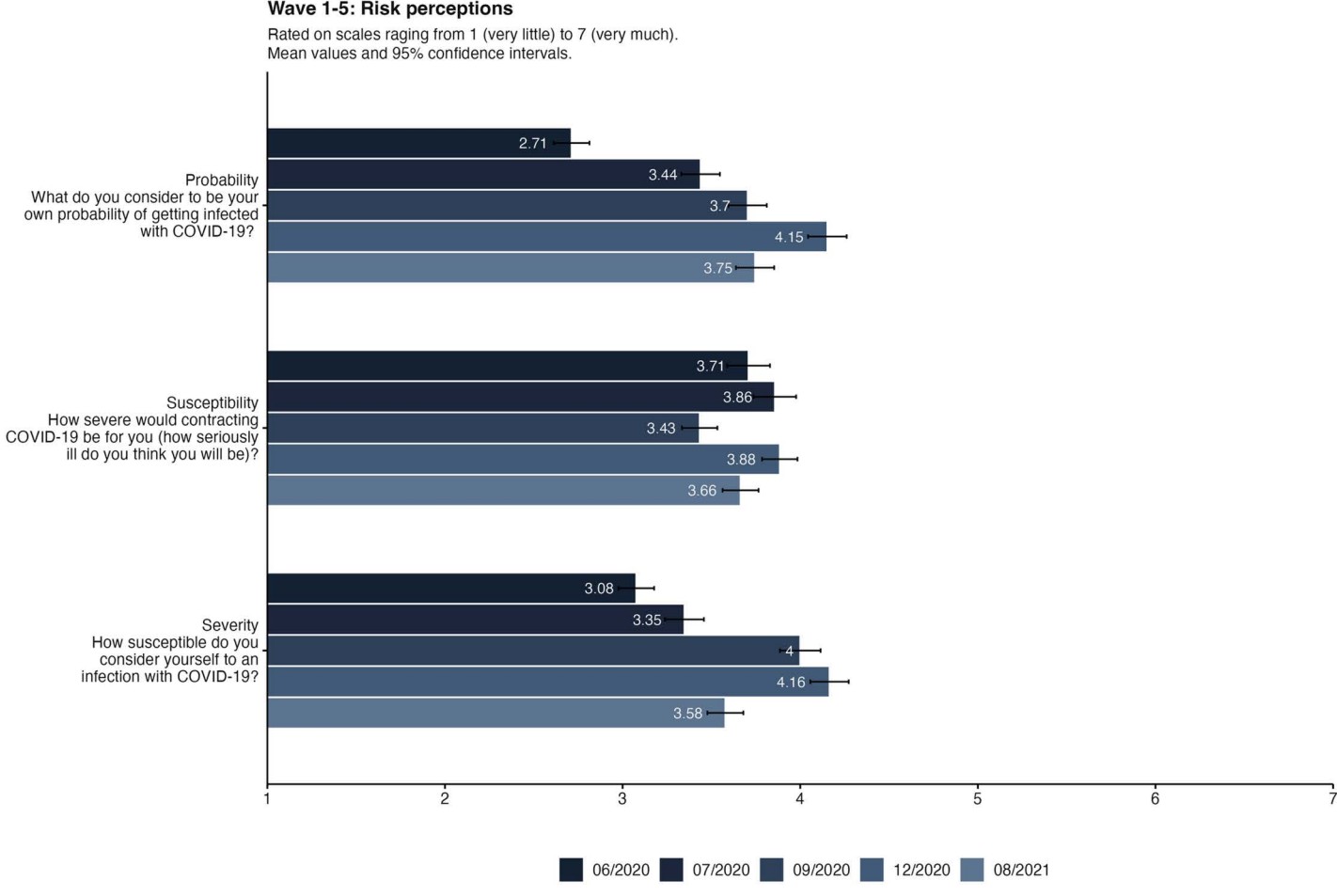

**Fig 2. Risk perception items in all five waves of data collection.** Bars displaying mean values and 95%-Confidence Intervals.

**Table 4. Analysis of variance and tukey post-hoc comparisons for emotional response in five waves of data collection.**

**ANOVA**

|  | df | Sum Sq | Mean Sq | F | p |
|---|---|---|---|---|---|
| Wave | 4 | 185 | 46.13 | 35.1 | **<.001** |
| Error | 5190 | 6819 | 1.31 |  |  |

Tukey Post-hoc comparisons

|  | | $M_{diff}$ | LCI | UCI | $P_{adj.}$ |
|---|---|---|---|---|---|
| **W2 – W1** |  | 0.260 | 0.122 | 0.398 | **<.001** |
| **W3 – W1** |  | 0.175 | 0.039 | 0.311 | **0.004** |
| **W4 – W1** |  | 0.575 | 0.439 | 0.710 | **<.001** |
| **W5 – W1** |  | 0.244 | 0.107 | 0.382 | **<.001** |
| **W3 – W2** |  | −0.085 | −0.222 | 0.053 | 0.449 |
| **W4 – W2** |  | 0.315 | 0.177 | 0.453 | **<.001** |
| **W5 – W2** |  | −0.015 | −0.155 | 0.124 | 0.998 |
| **W4 – W3** |  | 0.400 | 0.264 | 0.535 | **<.001** |
| **W5 – W3** |  | 0.069 | −0.068 | 0.207 | 0.643 |
| **W5 – W4** |  | −0.330 | −0.468 | −0.193 | **<.001** |

Results for the analysis of variance with survey wave as independent and emotional response (including eight items about fear, closeness, media hype, worry) as the dependent variable. Tukey Post-Hoc tests compare the means between the five waves (W1, W2, W3, W4, W5). Adjusted p-values and 95%-confidence intervals (LCI = lower confidence interval bound; UCI = upper confidence interval bound) show significant differences.

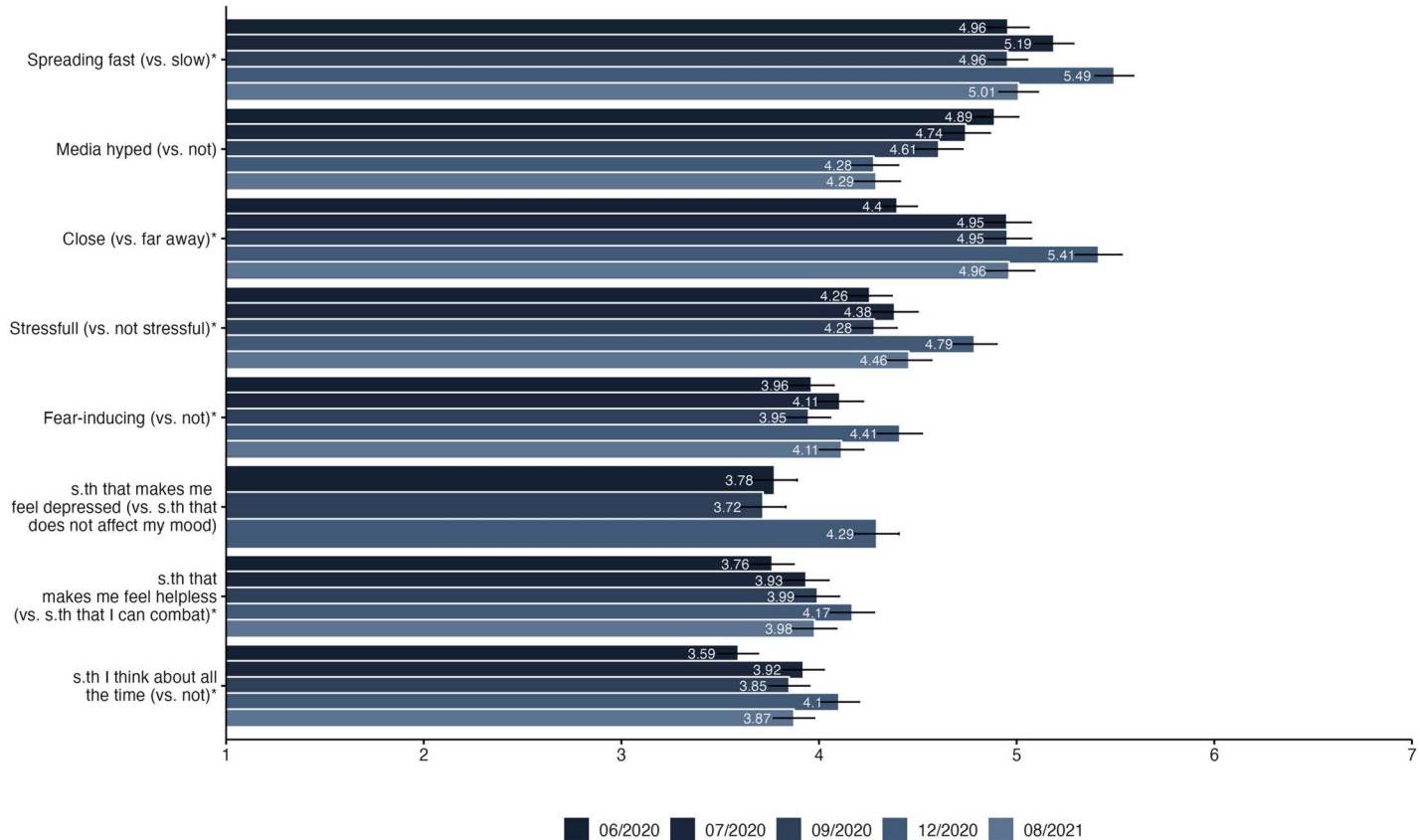

**Fig 3. Emotional response items in all five waves of data collection.** Emotional response to the pandemic was measured on 7-point semantic differentials. The items marked with asterisks are included in the emotional response scale. Bars displaying mean values and 95%-Confidence Intervals.

**Table 5. Analysis of variance and tukey post-hoc comparisons for trust in institutions in five waves of data collection.**

| ANOVA | | | | | |
|---|---|---|---|---|---|
| | df | Sum Sq | Mean Sq | F | p |
| Wave | 4 | 32 | 7.978 | 2.838 | .023 |
| Error | 5152 | 14486 | 2.812 | | |
| Tukey Post-hoc comparisons | | | | | |
| | | $M_{diff}$ | LCI | UCI | $P_{adj.}$ |
| **W2 – W1** | | −0.011 | −0.214 | 0.191 | 1.000 |
| **W3 – W1** | | −0.142 | −0.342 | 0.057 | 0.290 |
| **W4 – W1** | | −0.174 | −0.373 | 0.025 | 0.120 |
| **W5 – W1** | | 0.010 | −0.192 | 0.212 | 1.000 |
| **W3 – W2** | | −0.131 | −0.334 | 0.071 | 0.392 |
| **W4 – W2** | | −0.163 | −0.365 | 0.040 | 0.181 |
| **W5 – W2** | | 0.022 | −0.184 | 0.227 | 0.999 |
| **W4 – W3** | | −0.032 | −0.231 | 0.167 | 0.993 |
| **W5 – W3** | | 0.153 | −0.049 | 0.355 | 0.235 |
| **W5 – W4** | | 0.184 | −0.017 | 0.386 | 0.092 |

Results for the analysis of variance with survey wave as independent and trust (in the doctor, in the ministry of health, in the institute for public health) as the dependent variable. Tukey Post-Hoc tests compare the means between the five waves (W1, W2, W3, W4, W5). Adjusted p-values and 95%-confidence intervals (LCI = lower confidence interval bound; UCI = upper confidence interval bound) show significant differences.

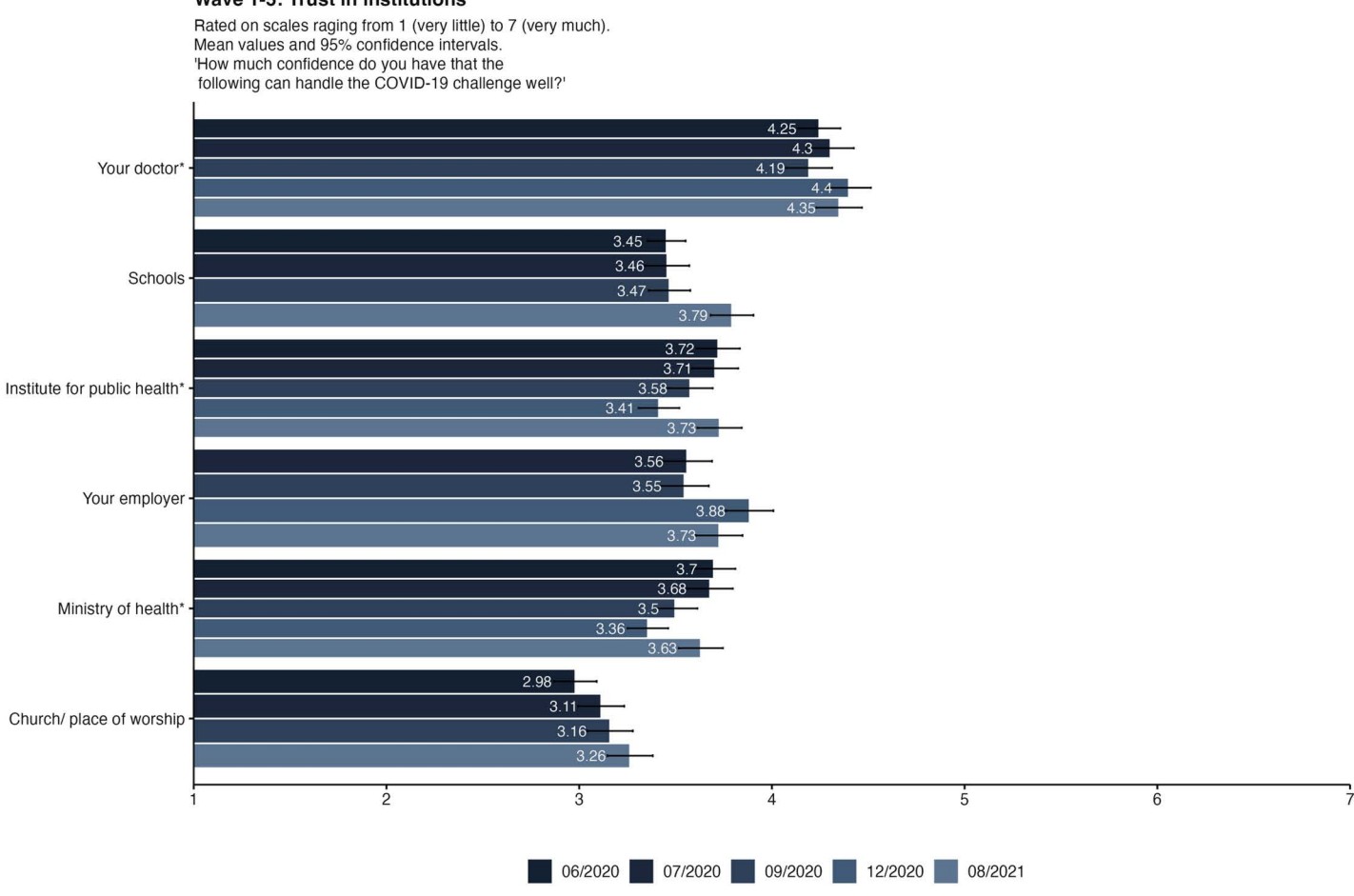

**Fig 4. Items for trust in institutions in all five waves of data collection.** The items marked with asterisks are included in the emotional response scale. Bars displaying mean values and 95%-Confidence Intervals.

**Table 6. Analysis of variance and tukey post-hoc comparisons for health literacy in four waves of data collection.**

**ANOVA**

|  | df | Sum Sq | Mean Sq | F | p |
|---|---|---|---|---|---|
| **Wave** | 3 | 10 | 3.452 | 1.818 | .142 |
| **Error** | 4137 | 7857 | 1.899 |  |  |

**Tukey Post-hoc comparisons**

|  |  | $M_{diff}$ | LCI | UCI | $P_{adj.}$ |
|---|---|---|---|---|---|
| **W3 – W2** |  | −0.094 | −0.250 | 0.062 | 0.411 |
| **W4 – W2** |  | 0.017 | −0.139 | 0.173 | 0.993 |
| **W5 – W2** |  | 0.035 | −0.123 | 0.193 | 0.942 |
| **W4 – W3** |  | 0.110 | −0.043 | 0.264 | 0.250 |
| **W5 – W3** |  | 0.129 | −0.027 | 0.284 | 0.146 |
| **W5 – W4** |  | 0.018 | −0.137 | 0.174 | 0.991 |

Results for the analysis of variance with survey wave as independent and trust (in the doctor, in the ministry of health, in the institute for public health) as the dependent variable. Tukey Post-Hoc tests compare the means between the four waves (W1, W2, W3, W4). Adjusted p-values and 95%-confidence intervals (LCI = lower confidence interval bound; UCI = upper confidence interval bound) show significant differences.

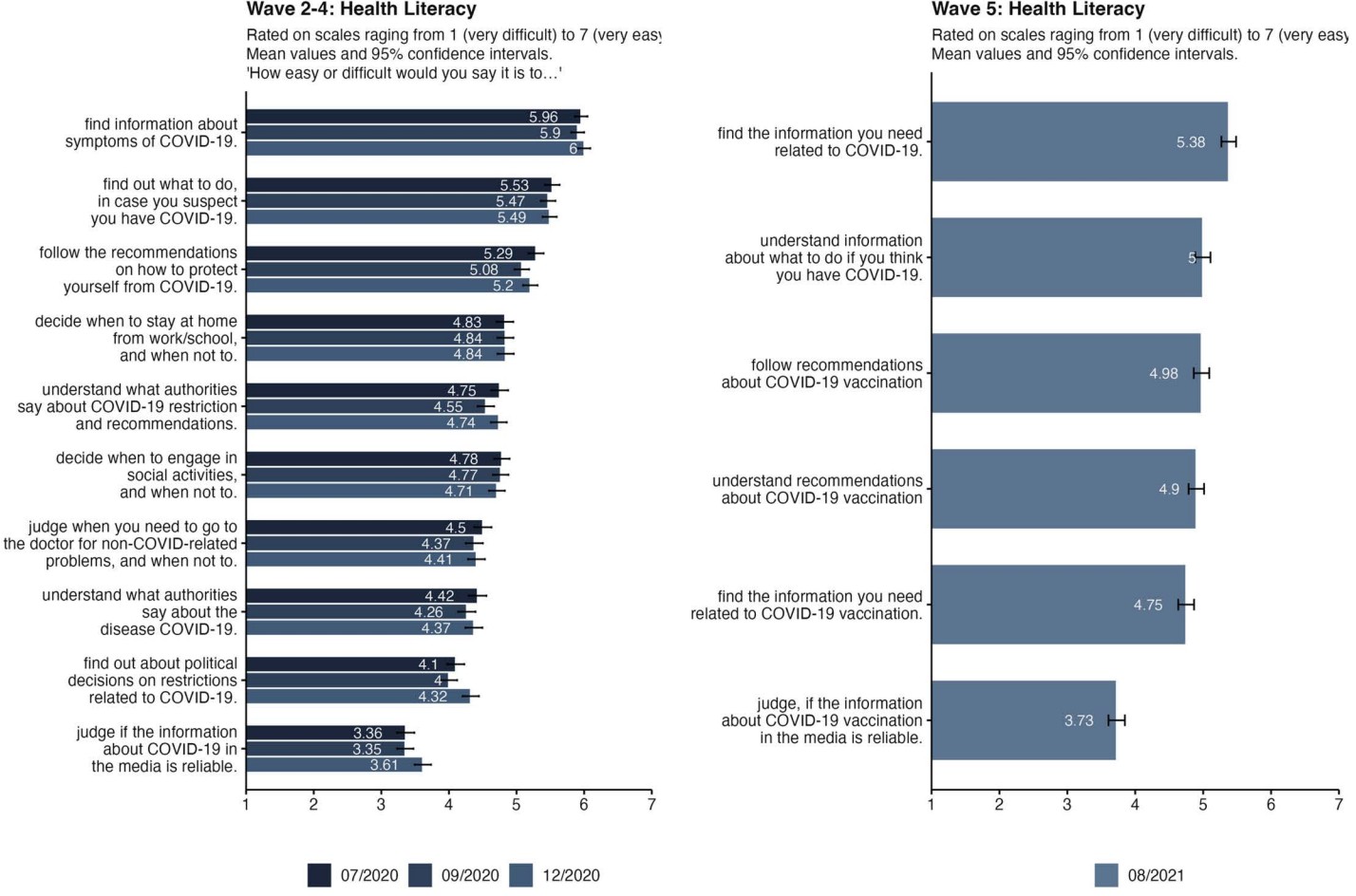

**Fig 5. Health literacy items in the original scale (left panel; wave 2–4) and the adapted scale (right panel; wave 5) including vaccination.** Bars displaying mean values and 95%-Confidence Intervals. The health literacy scale was adapted between wave 4 and wave 5., to cover additional questions about the vaccination.

**Table 7. Analysis of variance and Tukey post-hoc comparisons for fairness and policy acceptance in five waves of data collection.**

**ANOVA**

|  | df | Sum Sq | Mean Sq | F | p |
|---|---|---|---|---|---|
| Wave | 4 | 32 | 7.978 | 2.838 | .023 |
| Error | 5152 | 14486 | 2.812 |  |  |

Tukey Post-hoc comparisons

|  | $M_{diff}$ | LCI | UCI | $p_{adj.}$ |
|---|---|---|---|---|
| **W2 – W1** | −0.010 | −0.184 | 0.164 | 1.000 |
| **W3 – W1** | −0.365 | −0.536 | −0.193 | **<.001** |
| **W4 – W1** | −0.001 | −0.172 | 0.171 | 1.000 |
| **W5 – W1** | −0.200 | −0.373 | −0.026 | **0.015** |
| **W3 – W2** | −0.355 | −0.528 | −0.181 | **<.001** |
| **W4 – W2** | 0.009 | −0.164 | 0.183 | 1.000 |
| **W5 – W2** | −0.190 | −0.366 | −0.014 | **0.027** |
| **W4 – W3** | 0.364 | 0.193 | 0.535 | **<.001** |
| **W5 – W3** | 0.165 | −0.008 | 0.338 | 0.071 |
| **W5 – W4** | −0.199 | −0.372 | −0.026 | **0.015** |

Results for the analysis of variance with survey wave as independent and fairness/policy acceptance (three items: were decisions perceived as fair, right, or exaggerated) as the dependent variable. Tukey Post-Hoc tests compare the means between the five waves (W1, W2, W3, W4, W5). Adjusted p-values and 95%-confidence intervals (LCI = lower confidence interval bound; UCI = upper confidence interval bound) show significant differences.

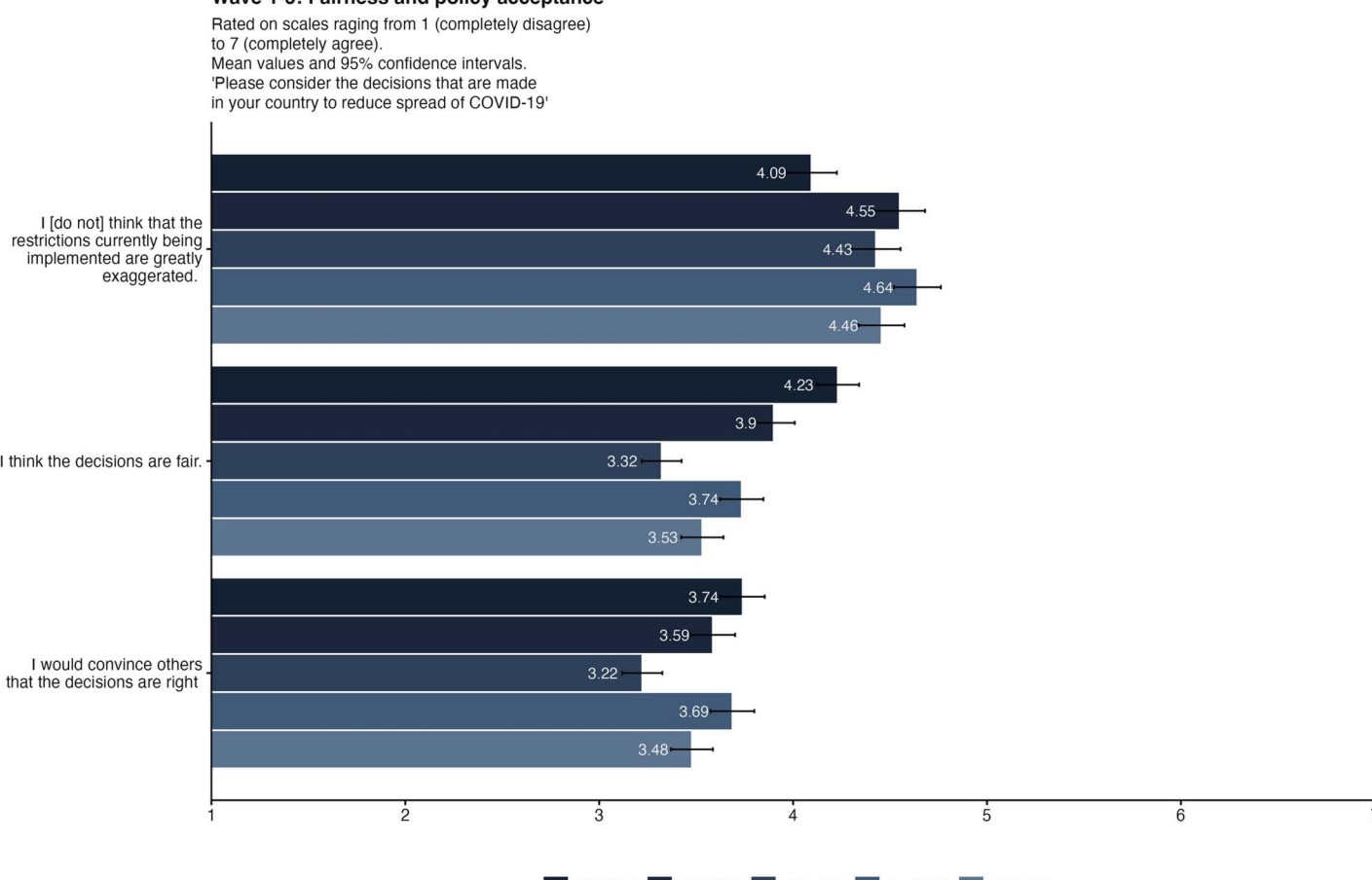

**Fig 6. Fairness and policy acceptance items in four waves of data collection.** Bars displaying mean values and 95%-Confidence Intervals. The first item was recoded, so that greater values always display greater acceptance. The original item was "I think, that the restrictions currently being implemented are greatly exaggerated".

**Table 8. Results for an analysis of variance and Tukey post-hoc tests to test, if protective behaviors change over time.**

**ANOVA**

|  | df | Sum Sq | Mean Sq | F | p |
|---|---|---|---|---|---|
| Wave | 4 | 12.71 | 3.176 | 63.57 | **<.001** |
| Error | 5185 | 259.07 | 0.050 |  |  |

Tukey Post-hoc comparisons

|  |  | $M_{diff}$ | LCI | UCI | $p_{adj.}$ |
|---|---|---|---|---|---|
| **W2 – W1** |  | −0.116 | −0.143 | −0.089 | **<.001** |
| **W3 – W1** |  | −0.116 | −0.142 | −0.089 | **<.001** |
| **W4 – W1** |  | −0.072 | −0.098 | −0.045 | **<.001** |
| **W5 – W1** |  | −0.139 | −0.166 | −0.112 | **<.001** |
| **W3 – W2** |  | 0.000 | −0.027 | 0.027 | 1.000 |
| **W4 – W2** |  | 0.044 | 0.017 | 0.071 | **<.001** |

*(Continued)*

**Table 8.** (Continued)

| ANOVA | | | | | |
|---|---|---|---|---|---|
| | df | Sum Sq | Mean Sq | F | p |
| **W5 – W2** | | −0.023 | −0.051 | 0.004 | 0.129 |
| **W4 – W3** | | 0.044 | 0.018 | 0.071 | **<.001** |
| **W5 – W3** | | −0.023 | −0.050 | 0.003 | 0.120 |
| **W5 – W4** | | −0.068 | −0.095 | −0.041 | **<.001** |

Results for the analysis of variance with survey wave as independent and protective behaviors (seven items, including washing hands, wearing a face mask, disinfecting surfaces, avoiding social events) as the dependent variable. Tukey Post-Hoc tests compare the means between the five waves (W1, W2, W3, W4, W5). Adjusted p-values and 95%-confidence intervals (LCI = lower confidence interval bound; UCI = upper confidence interval bound) show significant differences.

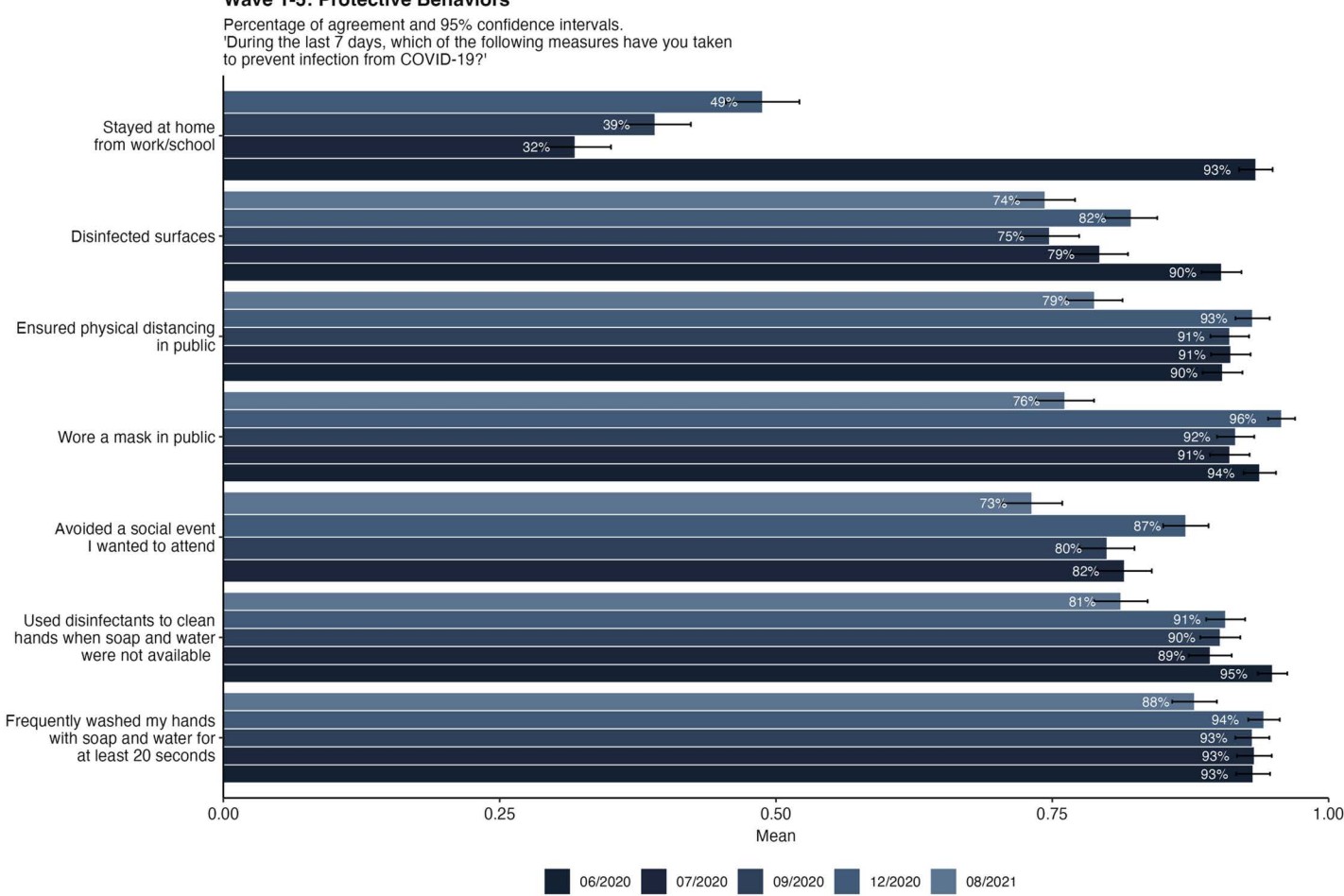

**Fig 7. Items for applying protective behaviors in all five waves of data collection.** Self-reported number of protective behaviors during the COVID-19 Pandemic in FBIH. Participants over the five waves of data collection were asked whether they performed the respective behavior during the last 7 days. In wave 5, the answer format was changed from a binary format (1 *yes*, 0 *no*) to a 7-point Likert scale (1 *never*, 7 *all of the time*). For comparability, the answers were recoded into binary format (1 – former scale points 5-7, 0 – former scale points 1-4).

**Table 9. Results for five linear regressions, predicting protective behavior by sociodemographic and behavioral insights between June 2020 and August 2021 in representative surveys in the Federation of Bosnia and Herzegovina.**

| Predictors | Wave 1 | | | Wave 2 | | | Wave 3 | | | Wave 4 | | | Wave 5 | | |
|---|---|---|---|---|---|---|---|---|---|---|---|---|---|---|---|
| | Beta | standardized CI | p | Beta | standardized CI | p | Beta | standardized CI | p | Beta | standardized CI | p | Beta | standardized CI | p |
| Age | 0.01 | −0.05 – 0.07 | .711 | 0.06 | 0.00 – 0.12 | **.046** | 0.11 | 0.06 – 0.17 | **<.001** | 0.08 | 0.03 – 0.14 | **0.004** | 0.12 | 0.06 – 0.18 | **<0.001** |
| Gender m (vs. f) | −0.17 | −0.28 – −0.05 | **.004** | −0.17 | −0.28 – −0.05 | **.004** | −0.16 | −0.27 – −0.05 | **.005** | −0.17 | −0.28 – −0.06 | **0.002** | −0.01 | −0.13 – 0.10 | 0.802 |
| Education high (vs. low) | −0.03 | −0.16 – 0.11 | .687 | 0.09 | −0.03 – 0.22 | .152 | 0.07 | −0.05 – 0.20 | .252 | 0.17 | 0.05 – 0.30 | **0.007** | −0.03 | −0.15 – 0.10 | 0.679 |
| Working in Healthcare | 0.04 | −0.08 – 0.15 | .521 | −0.02 | −0.23 – 0.19 | .836 | 0.12 | −0.07 – 0.30 | .206 | 0.17 | −0.02 – 0.36 | 0.082 | 0.12 | −0.08 – 0.32 | 0.230 |
| Own children (in their household) | 0.01 | −0.11 – 0.13 | .853 | 0.11 | −0.00 – 0.23 | .050 | 0.01 | −0.10 – 0.12 | .863 | 0.02 | −0.09 – 0.13 | 0.721 | −0.06 | −0.18 – 0.06 | 0.327 |
| Living urban (vs. rural) | 0.17 | 0.06 – 0.28 | **.004** | 0.07 | −0.05 – 0.18 | .257 | 0.02 | −0.10 – 0.14 | .736 | 0.10 | −0.01 – 0.22 | 0.071 | 0.11 | −0.00 – 0.22 | 0.061 |
| Emotional response | 0.11 | 0.05 – 0.17 | **<.001** | 0.20 | 0.13 – 0.26 | **<.001** | 0.16 | 0.10 – 0.22 | **<.001** | 0.21 | 0.15 – 0.27 | **<0.001** | 0.13 | 0.07 – 0.19 | **<0.001** |
| Trust | 0.19 | 0.13 – 0.25 | **<.001** | 0.14 | 0.07 – 0.20 | **<.001** | 0.06 | −0.00 – 0.13 | .051 | 0.09 | 0.03 – 0.16 | **0.005** | 0.21 | 0.14 – 0.27 | **<0.001** |
| Risk perception | 0.06 | −0.00 – 0.12 | .050 | 0.09 | 0.02 – 0.15 | **.008** | 0.13 | 0.07 – 0.19 | **<.001** | 0.10 | 0.04 – 0.16 | **0.002** | 0.18 | 0.12 – 0.24 | **<0.001** |
| Policy acceptance | 0.13 | 0.07 – 0.20 | **<.001** | 0.12 | 0.05 – 0.19 | **.001** | 0.20 | 0.13 – 0.26 | **<.001** | 0.16 | 0.10 – 0.23 | **<0.001** | 0.09 | 0.02 – 0.15 | **0.009** |
| Self-efficacy | 0.05 | −0.01 – 0.11 | .099 | 0.07 | 0.01 – 0.13 | **.024** | 0.01 | −0.05 – 0.06 | .862 | 0.07 | 0.01 – 0.13 | **0.016** | | | |
| Health literacy | | | | 0.10 | 0.04 – 0.16 | **.002** | 0.09 | 0.03 – 0.15 | **.004** | 0.03 | −0.03 – 0.10 | 0.285 | 0.04 | −0.02 – 0.10 | 0.181 |
| Observations | 1052 | | | 989 | | | 1058 | | | 1055 | | | 999 | | |
| R² / R² adjusted | 0.134/ 0.125 | | | 0.209/ 0.199 | | | 0.201/ 0.192 | | | 0.203/ 0.194 | | | 0.198/ 0.189 | | |

Beta values and standardized 95%-confidence intervals (CI) for predictors of protective behaviors. Seven Protective behaviors, such as washing hands, wearing a face mask, disinfecting surfaces and avoiding social events were included in a mean score, reaching from 0 (no measures applied) to 1 (all measures applied)

time. Table 6 shows that health literacy as a construct seems to be quite stable over the course of the pandemic. Looking into the descriptive differences of the single items in Fig 5 reveals great differences between respondents' perceived competence in different domains. Whereas most respondents were sure that they could find information about COVID-19 symptoms or vaccinations, far fewer were confident in their ability to judge reliability of information in the media.

## Fairness and policy acceptance

Participants in all waves were asked whether they accepted the official pandemic restrictions and if they perceived the restrictions as fair. An ANOVA revealed that these fairness and acceptance perceptions changed over time, as Table 7 shows. Yet we can see from Fig 6 that acceptance and fairness perceptions changed differently over the course of the pandemic. In all five waves, most respondents did not think of PHSM as exaggerated. However, fairness perceptions decreased over time, and being confident that the political decisions were right fluctuated as well. All measures significantly decreased during wave 3 in September 2020.

## Protective behaviors

Table 8 shows the results of a one-way ANOVA with the data collection wave as the independent variable and the mean value for protective behaviors as the dependent variable. Overall, protective behaviors decreased between June 2020 and August 2021. Post-hoc tests revealed that protective behavior was highest in June 2020, significantly increased again in the Winter of 2020 and then decreased to the lowest level in August 2021. Single descriptive developments for protective behaviors can be seen in Fig 7. Levels of handwashing, mask wearing, and physical distancing in public remained relatively high across the survey waves. People were less engaged with the disinfection of surfaces and avoiding social events. Staying home from work/school was the least observed protective behavior. Being female was associated with higher odds of protective behavior for most outcomes. Exceptions were wearing face masks and adapting their work situation. Associations between respondents' age and individual behavior change were inconsistent and mostly weak. In our case, education level was connected with better information about symptoms.

Antecedents of protective behavior are mostly robust over the different data collection waves, as can be seen from Table 9 showing all regression results. In all five regression analyses, higher emotional response and higher policy acceptance are associated with more protective behavior. In three to four of the regression analyses, higher age, being female (vs. male), having higher trust in institutions, and higher risk perception is associated with more protection behaviors. Less robust are the results for effects of education, urban vs. rural living, self-efficacy, and health literacy. Overall fit of the regressions is between moderate to high (Wave 1: $R^2 = .134$, Wave 2: $R^2 = .208$, Wave 3: $R^2 = .207$, Wave 4: $R^2 = .198$, Wave 5: $R^2 = .198$).

## Discussion

This survey and its accompanying data analysis illuminate the significant impact of behavioral determinants in relation to PHSM. For this summary discussion we will focus on four factors most strongly associated with protective behaviors: emotional response, risk perception, policy acceptance, and trust in institutions. We will also address health literacy and relate our findings to the broader research literature.

Emotional response was a strong predictor of protective behaviors. Our survey respondents reported the strongest emotional response to the perceived speed of the virus's spread and its nearness to them; they were less concerned about feeling helpless and thinking about the pandemic all the time. The perception of COVID-19 risks among the public and their

willingness to adhere to preventive measures, such as vaccination, in our study is influenced by various factors. Risk perception generally remains high; however, factors such as trust in governmental initiatives, a shared sense of responsibility, and active participation in health-protective behaviors can significantly shape and potentially reduce these perceptions [27]. The spike in emotional response in the fourth wave likely relates to heightened risk perceptions, which would be consistent with findings from other research (see below). Our findings on the predictive power of emotional responses align with broader literature from Europe, Asia and North America [28–33]. In European countries, perceived threat and negative emotional reactions, such as worry, have been consistently identified as key predictors of self-reported preventive behaviors during the COVID-19 pandemic [30]. Research shows that individuals perceiving greater threats and experiencing stronger negative emotions, including fear and anxiety, are more likely to adopt protective measures such as social distancing, mask-wearing, and adherence to health guidelines [29]. Kim et al. found specifically that *negative* emotions were strong predictors of protective behaviors in Korea [32]. Similarly, in a study by Liu et al., negative emotions increased risk perception, which in turn increased intentions to wear face masks in the US [33].

Regarding COVID-19 risk perception, respondents initially exhibited a noticeable increase, which continued to rise steadily through the first four waves of data collection, culminating in a significant surge during the fourth wave. In the Federation of Bosnia and Herzegovina (FBiH), where the pandemic was marked by fluctuating case numbers, the population may have experienced fatigue in consistently adhering to preventive measures [27]. The data indicate a pronounced increase in risk perception around weeks 44 to 49 in 2020 (corresponding with the fourth wave), which coincided with a high number of COVID-19 positive people and deaths [11]. Notably, individuals living in urban areas or dealing with chronic illnesses tended to demonstrate heightened levels of risk perception, emphasizing the influence of environmental and health-related factors on perceived risk intensity. Our analysis revealed a positive correlation between risk perceptions and the increased adoption of protective behaviors, suggesting that heightened risk perception promotes the adoption of protective measures.

Our findings reinforce trends observed in national and international studies. Multiple studies [34–39] have explored risk perceptions and consistently concluded that individuals perceiving greater risks are more likely to adopt protective behaviors. In another study in the FBIH, Musa et al. conducted a comprehensive assessment of risk perception, trust in institutions, and the impact on vaccine intention [40]. The index of risk perception—encompassing self-assessed probability of COVID-19 infection, susceptibility and perceived severity of COVID-19 illness—emerged as a highly significant driver of positive vaccine intentions. Noteworthy risk factors, such as older age, underlying medical conditions, and occupation requiring close contact with COVID-19-positive patients were identified as contributors to risk perception [40,41]. In a study conducted in Italy, Di Giuseppe et al. found a high risk awareness among the study population, though respondents had low confidence in their ability to protect themselves from infection [42]. These findings correspond with our results, where knowledge about protective measures was higher than behavioral self-efficacy across all data collections.

Policy acceptance and perceptions of fairness were also important predictors of protective behaviors in our analysis. Though participants did not generally think that PHSM were exaggerated, they had dimmer views of fairness, and relatively few said they would try to convince others that PHSM decisions were right. The lower ratings on these latter two items suggest low confidence in decision-making institutions. This conjecture is supported by the wider research literature, where policy acceptance is typically studied in the context of trust, whether in government or science [43,44]. For instance, Guglielmi et al. [45] show how institutional

confidence and specific political support influence public acceptability of containment measures in Italy, suggesting that trust in institutions can either facilitate or hinder compliance depending on political contexts. In the FBIH pandemic monitoring here, we can see that the values for trust and policy fairness decreased to the same amount from wave 1 to wave 3, but we do not see that their increase is parallel in the last 2 waves. This evidence indeed points to trust as playing an important (albeit complex) mediating role in policy acceptance [45,46].

Though our surveys did not measure trust in science, they showed that trust in institutions was generally low across all waves and declined for the Ministry of Health and the Institute for Public Health during the fall and winter of 2020. However, trust in family doctors remained relatively high and was linked to protective behaviors. Trust in government has generally been found to be positively associated with protective behaviors [47–49], though sometimes with a moderate impact in modelling studies [50]. Similar to a study from the Netherlands regarding H1N1 [51], our results have shown that people who trust the health ministry and the media are more likely to adopt recommended behavior to control the spread of a pandemic virus, compared with people who lack trust in the health ministry and the media.

Health literacy was not a significant antecedent of predictive behaviors in our analysis. Although health literacy is repeatedly associated in the literature with people being better able to understand and apply more proactive health protection, there is no relationship between health literacy and PHSM in the multiple regression. The global pandemic and the associated media attention could have made it easier to find information, so that everyone had access to easily understandable information. In line with this explanation, the data for health literacy showed that information seeking was assessed as relatively straightforward, with individuals feeling capable of finding information about symptoms or regulations. In contrast, lower values were evident when it came to translating this information into daily life, indicating difficulties in incorporating regulations and recommendations into everyday decisions

Finally, it remains to mention the limitations of these pandemic monitoring surveys in the FBIH. Online samples hold a potential risk of bias because they reach only the people who have access to the internet. Even if there is a high level of internet usage (73.2% in 2020, 75.7% in 2021, and 78.7% in 2022 [52], this selection bias might be higher in older generations, so all effects where age is relevant would need to be replicated. Additional telephone surveys or household interviews might increase further insights about the effect size of this sampling bias and include underrepresented groups in the data set. Even if this additional sampling strategy was no option for this monitoring survey, future monitorings could include mixed methods in sampling. On the other hand, advantages of online surveys included that the samples were drawn to be representative and were collected in a short time period, reaching participants all over the country. Rapid data collection was of high importance during the pandemic to prevent additional confounding due to fast-changing measures and recommendations. Further, online samples are known to produce less social desirability biases [53], and especially when strict measures and recommendations were in place, social desirability might have biased the responses in interview formats. As a further limitation, we experienced a difference in the invitation/ participation ratio over the waves, with more people needed to be invited at the earlier waves of the cross-sectional survey. This might be caused by the intense information density at the earlier stages of the pandemic. In consequence, people may have been striving more for distraction, producing a self-selection bias in the early waves of our data. The people who were affected most strongly by the pandemic may have evaded the questionnaire study. Future pandemic studies with special panels could carefully address precisely these people, e.g., with help hotlines or counseling services, including qualitative approaches alongside the quantitative data for the general population. With this approach, behavioral insights could also be used for this target group to adapt pandemic decisions and support even better to the people who are psychologically affected most. Despite these limitations, the data

collected were robust enough to contribute directly to the COVID-19 response in the FBIH in communications and messaging, interventions, programs, and policy.

## Conclusion

This study highlights the importance of "public pulse monitoring" through ongoing surveys which can help tailor communication about PHSM to increase compliance. Our surveys in the FBIH showed how pandemic perceptions changed over time, and that responses to PHSM were influenced particularly by trust in institutions, risk perceptions, and emotional response. Given that our findings generally accord with similar conclusions from the wider research literature, we can offer several recommendations for future pandemic preparedness and to increase societies' resilience in health emergencies.

First, the importance of trust is paramount; it is something that authorities must strive for and promote. Because public trust has a strong association with compliance to PHSM, and is vulnerable to eroding over time, there is a need to regularly review communication approaches and tailor information strategies for specific groups to help improve protective behaviors. Which channels and trusted sources may work best should be investigated continuously in behavioral insights surveys, as trusted sources and channels (such as social media platforms) change over time.

Second, and likewise related to communication, our results underscore the crucial role of behavioral insights surveys in providing well-tested and evidence-based information about risks and PHSM. Our data show how individuals with heightened risk perceptions were more inclined to adopt preventive measures and cautious behaviors, especially in the later stages of the pandemic. Hence, public health/scientific leadership will be essential in crafting effective risk communication strategies for future health emergencies [54]. Emergency recommendations should undergo thorough assessments for comprehensibility. Health messaging should also include clearly defined actions and provide practical examples to illustrate the recommendations.

Third, and following from the previous two points, our analysis shows that communication about PHSM and promoting healthy behaviors, is not necessarily "one size fits all." Behavioral surveys can provide valuable insights into how different sectors of a population will respond differently to emergency recommendations. For example, one possible explanation for divergent responses to PHSM we observed is that some measures were abandoned earlier when they entailed higher social and financial consequences for particular groups. Therefore, on the basis of behavioral research, emergency communication should address such consequences, act supportively, and if necessary, consider monetary or social assistance to particularly affected target groups.

## Supporting Information

**S1 Appendix.  The appendix contains seven tables, including regressions for risk perceptions, self-efficacy, trust in institutions, emotional pandemic response, health literacy and policy acceptance.** It also includes descriptive gender differences of self-reported pandemic behaviors.
(DOCX)

## Acknowledgments

The initiative to collect these data as pandemic emergency response was led by the Behavioural and Cultural Insights Unit (BCI) in close collaboration with the regional WHO Europe emergencies programme, including the Risk Communication and Community

Engagement (RCCE), the Public Health and Social Measures (PHSM) and the Essential Health Services (EHS) pillars.

**Disclaimer:** The authors affiliated with the World Health Organization (WHO) are alone responsible for the views expressed in this publication and they do not necessarily represent the decisions or policies of the WHO.

## Author contributions

**Conceptualization:** Šeila Cilović-Lagarija, Sarah Eitze, Siniša Skočibušić, Sanjin Musa, Stela Stojisavljević, Faris Dizdar, Mirza Palo, Dorit Nitzan, Miguel Telo de Arriaga, Martha Scherzer, Katrine Bach Habersaat.

**Data curation:** Sarah Eitze, Haris Šabanović, Faris Dizdar, Martha Scherzer.

**Formal analysis:** Sarah Eitze.

**Funding acquisition:** Šeila Cilović-Lagarija, Faris Dizdar, Mirza Palo, Martha Scherzer, Katrine Bach Habersaat.

**Investigation:** Šeila Cilović-Lagarija, Sarah Eitze, Stela Stojisavljević, Dorit Nitzan, Miguel Telo de Arriaga, Martha Scherzer, Katrine Bach Habersaat.

**Methodology:** Sarah Eitze, Siniša Skočibušić, Dorit Nitzan, Martha Scherzer.

**Project administration:** Šeila Cilović-Lagarija, Stela Stojisavljević, Haris Šabanović, Faris Dizdar, Mirza Palo, Martha Scherzer, Katrine Bach Habersaat.

**Resources:** Siniša Skočibušić, Sanjin Musa, Stela Stojisavljević, Mirza Palo, Katrine Bach Habersaat.

**Supervision:** Šeila Cilović-Lagarija, Siniša Skočibušić, Sanjin Musa, Faris Dizdar, Dorit Nitzan, Miguel Telo de Arriaga, Martha Scherzer, Katrine Bach Habersaat.

**Validation:** Haris Šabanović, Benjamin Curtis.

**Visualization:** Sarah Eitze, Benjamin Curtis.

**Writing – original draft:** Šeila Cilović-Lagarija, Sarah Eitze, Martha Scherzer, Benjamin Curtis, Katrine Bach Habersaat.

**Writing – review & editing:** Šeila Cilović-Lagarija, Sarah Eitze, Siniša Skočibušić, Sanjin Musa, Stela Stojisavljević, Haris Šabanović, Faris Dizdar, Mirza Palo, Dorit Nitzan, Miguel Telo de Arriaga, Martha Scherzer, Katrine Bach Habersaat.

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
