## [Decision Letter · Decision Letter 0]

1 Dec 2024

PONE-D-24-28827Behavioral insights during the COVID-19 pandemic in the Federation of Bosnia and Herzegovina: the role of trust, health literacy, risk and fairness perceptions in compliance with public health and social measuresPLOS ONE

Dear Dr. Eitze,

Thank you for submitting your manuscript to PLOS ONE. After careful consideration, we feel that it has merit but does not fully meet PLOS ONE’s publication criteria as it currently stands. Therefore, we invite you to submit a revised version of the manuscript that addresses the points raised during the review process.

**I would like to see your paper published but I really need you to go thoroughly through all the comments reviewers gave you. Especially the very detailed observations that Reviewer 1 is giving you. Further, I need you to answer each of reviewers' comments and to address the changes made following the comments in the manuscript.**

**I strongly recommend to comply with Reviewer 1 provided solutions or to answer in details why something is not possible to accept.**

We look forward to receiving your revised manuscript.

Kind regards,

Iskra Alexandra Nola

Academic Editor

PLOS ONE

**Journal Requirements:**

4. Please upload a new copy of Figure 5 as the detail is not clear. Please follow the link for more information: https://blogs.plos.org/plos/2019/06/looking-good-tips-for-creating-your-plos-figures-graphics/" https://blogs.plos.org/plos/2019/06/looking-good-tips-for-creating-your-plos-figures-graphics/"

Reviewers' comments:

Reviewer's Responses to Questions

**Comments to the Author**

1. Is the manuscript technically sound, and do the data support the conclusions?

Reviewer #1: No

Reviewer #2: Yes

2. Has the statistical analysis been performed appropriately and rigorously? 

Reviewer #1: Yes

Reviewer #2: Yes

3. Have the authors made all data underlying the findings in their manuscript fully available?

Reviewer #1: Yes

Reviewer #2: Yes

4. Is the manuscript presented in an intelligible fashion and written in standard English?

Reviewer #1: No

Reviewer #2: Yes

5. Review Comments to the Author

**Reviewer #1:**  Dear Authors:

kindly address the follwoing:

Abstarct:The transition between sections could be smoother.

- Consider reordering phrases like: "The present study provides behavioral insights about self-reported compliance with PHSM" could be simplified to "This study examines self-reported compliance with PHSM," reducing wordiness.

- The description of the sample size ("1000 people representative of FBIH...") could mention how the representativeness was ensured. Did you use specific quotas for age, sex, and education level?

- You might also want to clarify what "cross-sectional studies" means in this context. Was data collection repeated in different periods (longitudinal cross-sectional design), or was it one cross-sectional study with waves?

- Avoid ambiguous terms like "key drivers" unless they are defined explicitly.

-It would help to specify which variables were strongest in the regression model and provide some context for the strength of associations (e.g., R-squared values)

- The suggestion to support health literacy in specific groups could benefit from specifying what groups were most vulnerable or poorly informed based on the findings.

Introduction:

- The opening sentence could be clearer and more engaging. Instead of starting with “Behavioral insights during the COVID-19 pandemic 60 in the Federation of Bosnia and Herzegovina,” consider starting with a broader statement about the importance of understanding population behaviors during health crises.

-Clarity in Purpose:

The introduction could benefit from explicitly stating early on why understanding these behaviors matters for the success of PHSM, tying this more closely to the study being introduced.

Suggested revision: This study aimed to assess the behavioral determinants that influence compliance with PHSM in the Federation of Bosnia and Herzegovina (FBIH) during the COVID-19 pandemic.

- The transition from the general discussion of PHSM to Bosnia’s specific context could be made smoother.

- The mention of the WHO PHSM recommendations and the survey tool implementation in Bosnia appears twice. It could be streamlined to avoid repetition.

Methods:

The sentence starting with "Time of data collection" (lines 96-98) is a bit awkward and can be restructured for better readability. It might flow better if the dates for each wave are listed in a tabular format or set apart as bullet points.

-The response rate section (lines 102-105) is informative but could be made clearer with some rewording. The phrasing feels slightly redundant and could use more precision in presenting the varying rates.

- The sentence "Dropout rate for all five waves were 8%" (line 109) should have more context on what this means for the study. Does this reflect a good retention rate? Additionally, "were" should be corrected to "was."

- It would help to clarify why there was such a variation in the number of people invited to each wave and the response rates. It might indicate something about the survey methodology or external factors.

- Remove unnecessary details: For example, in the "Health literacy" section, you mention internal consistency in both waves, but this could be summarized without repeating the Cronbach's alpha scores twice.

- Avoid over-explaining Likert scales: In multiple instances, the explanation of Likert scales (1 to 7) could be simplified. After the first explanation, the reader should understand how the scale works, so reiterating it for every variable is unnecessary.

- Separate statistical analysis section more clearly: While you do have a statistical analysis section (line 161), the explanations of statistical tests used, like ANOVA and multiple regression, are a bit disjointed. It might help to structure this as a formal, separate subsection that outlines the methodology in a clearer step-by-step manner.

- Explain variable selection more clearly: You mention why certain items were excluded from certain scales (e.g., in the Emotional Response section). This could be more explicit, explaining the reasoning behind why a low correlation or Cronbach's alpha led to their removal. Additionally, why you chose to only focus on these specific seven variables should be justified more thoroughly in the introduction to this section.

- Explain the statistical tests more clearly: While you describe using one-way ANOVA and Tukey post-hoc tests, it could be helpful to include brief reasoning on why these tests were chosen (i.e., for comparing means across multiple groups) and how they were applied in the context of the study.

-Discussion:

-The paragraph discussing health literacy (lines 412-423) could benefit from a more explicit connection between the data and its implications. You mention it was not a significant antecedent, but then delve into the details without clearly tying back to this point. A stronger conclusion for this section would enhance the overall coherence.

-Some sections, like the one on emotional response (lines 367-372), could use more precise connections between your data and the broader literature. You cite multiple studies, but a clearer explanation of how your results align or diverge from these findings could strengthen the analysis. For example, expand on how the spike in emotional response in the fourth wave specifically impacted protective behaviors in comparison to other waves.

- While you effectively reference several studies, you could integrate them more dynamically into your discussion. Rather than listing studies as standalone facts, try weaving their conclusions into your argument.

- The limitations section is well-acknowledged, but you could go deeper into how these limitations might have influenced your results. For instance, what specific elements of trust or health literacy might have been underrepresented due to sampling bias (lines 424-428)? Offering suggestions for future research or alternative methodologies could enhance the critical depth of this section.

- The discussion of policy acceptance and trust (lines 397-411) is insightful, but could benefit from more elaboration on the “complex mediating role” of trust. How does this complexity play out in your findings? Could more specific examples be provided from the survey data?

**Reviewer #2: ** The author used very simple English and the narrative of the paper is explicit with facts.

The methodology was appropriate for the study.

The statistical analyses helped in controlling for some cofounders and biases that could have arisen from dropout of

some study participants.

The author stated limitations to the study and this guides in interpretation of the results with caution especially in relation to age and access to internet.

6. PLOS authors have the option to publish the peer review history of their article (what does this mean? ). If published, this will include your full peer review and any attached files.

**Do you want your identity to be public for this peer review?** For information about this choice, including consent withdrawal, please see our Privacy Policy .

Reviewer #1: No

Reviewer #2: No

---

## [Author Response · Author response to Decision Letter 0]

21 Jan 2025

For a formatted version with highlighted changes in the paraghraphs, please see attached file "2024_PHSM_Response_R1.docx"

Dear Editors, dear Reviewers,

We would like to express our sincere gratitude for your invaluable feedback and suggestions on our manuscript. Your thorough review and insightful comments have significantly contributed to improving the clarity and overall quality of our work.

We are pleased to inform you that we have addressed all of your recommendations in the revised manuscript. The detailed responses to each comment are displayed in a point-by-point response below.

We believe the revisions we made have significantly enhanced the manuscript. The improvements have clarified the key points, refined the methodology, and strengthened the overall argument. We hope that the revised manuscript is now suitable for publication and look forward to your positive response.

Reviewer #1:

Abstract:

The transition between sections could be smoother.

>>We included some changes in the abstract, including the shift and integration of the last sentence from the background section to the materials and methods section, connecting the two sections by re-acknowledging the studies goals:

“This study examines self-reported compliance with PHSM during the COVID-19 pandemic in the Federation of Bosnia and Herzegovina (FBIH).

Materials and methods: We analyze the association between compliance and behavioral determinants, using data from five cross-sectional surveys that were conducted between June 2020 and August 2021 in FBIH. Data from 1000 people per wave were collected, with participants being representative of FBIH regarding age, sex, and education level based on the data from the latest census in Bosnia and Herzegovina. (p. 2, l.39-46)

- Consider reordering phrases like: "The present study provides behavioral insights about self-reported compliance with PHSM" could be simplified to "This study examines self-reported compliance with PHSM," reducing wordiness.

>> We changed the respective sentence in accordance with the reviewer’s suggestion:

“This study examines self-reported compliance with PHSM during the COVID-19 pandemic in the Federation of Bosnia and Herzegovina (FBIH).” (p. 2, l. 39-41)

- The description of the sample size ("1000 people representative of FBIH...") could mention how the representativeness was ensured. Did you use specific quotas for age, sex, and education level?

>> We specified which census data was used for quota-based sampling:

“We analyze the association between compliance and behavioral determinants, using data from five cross-sectional surveys that were conducted between June 2020 and August 2021 in FBIH. Quota-based sampling ensured that the 1000 people per wave were population representative regarding age, sex, and education level based on the data from the latest census in Bosnia and Herzegovina. “ (p. 2, l. 42-41)

- You might also want to clarify what "cross-sectional studies" means in this context. Was data collection repeated in different periods (longitudinal cross-sectional design), or was it one cross-sectional study with waves?

>> We included the information that 1000 people per wave were included and hope, to the limited word-count in the abstract, that this clarifies the data collection method.

For the respective sentence, please find the changes in the reference above.

- Avoid ambiguous terms like "key drivers" unless they are defined explicitly.

-It would help to specify which variables were strongest in the regression model and provide some context for the strength of associations (e.g., R-squared values)

>> We changed key drivers to significant predictors in the abstract. We also included the minimum and maximum standardized beta-weights from the five regression analyses behind every predictor to indicate minimum and maximum explanatory power of the predictors:

“In five wave-specific regression analyses, emotional response (βmin/max = .11*/.21*), risk perception (βmin/max = .06/.18*), policy acceptance (βmin/max = .09/.20*), and trust in institutions (βmin/max = .06/.21*) emerged as significant predictors of PHSM.“ (p. 2, l.53-55)

- The suggestion to support health literacy in specific groups could benefit from specifying what groups were most vulnerable or poorly informed based on the findings.

>> Due to the world limit for abstracts we needed to shorten the abstract, and we decided for a general recommendation for health literacy support here:

“The study also affirms the impact of public trust on compliance, the risk of eroding compliance over time, and the need for health literacy support to help reinforce protective behaviors. (p. 2, l.59)

Introduction:

- The opening sentence could be clearer and more engaging. Instead of starting with “Behavioral insights during the COVID-19 pandemic 60 in the Federation of Bosnia and Herzegovina,” consider starting with a broader statement about the importance of understanding population behaviors during health crises.

>> As we formatted our manuscript following the APA 7 guidelines, we included the Title of the manuscript again as title for the introduction. However, as we re-read the guidelines for PLOS-One, we see that other headlines are allowed as well. We thank the reviewer #1 for this recommendation, and chose the title “Knowing How, Knowing Why: A behavioral perspective on public health and social measures” for our introduction. (p. 3, l.61)

-Clarity in Purpose:

The introduction could benefit from explicitly stating early on why understanding these behaviors matters for the success of PHSM, tying this more closely to the study being introduced.

Suggested revision: This study aimed to assess the behavioral determinants that influence compliance with PHSM in the Federation of Bosnia and Herzegovina (FBIH) during the COVID-19 pandemic.

>> We included the general aim of this study at the end of the first paragraph of the introduction. We added the time frame of data collection, because we did not analyze data from following years.

“Risk perception and behaviors are essential for understanding compliance with PHSM, but in turn, they are influenced by several other factors, including individuals’ perceptions of the consistency, competence, fairness, and objectivity of governmental authorities (4,5,6). This study aimed to assess the behavioral determinants that influence compliance with PHSM in the Federation of Bosnia and Herzegovina (FBIH) during the first year of the COVID-19 pandemic 2020-2021. (p. 3, l.68-70)

- The transition from the general discussion of PHSM to Bosnia’s specific context could be made smoother.

>> We now included the beforementioned concepts of PHSM policies and behavioral research before introducing the studies context in the introduction:

“Both PHSM policies, such as cancellation of mass gatherings and complementary behavioral research monitorings were implemented in the Federation of Bosnia and Herzegovina (FBIH), tailored and adapted to the specific epidemiological situation that changed over the course of the pandemic (9,10,11). (p 3, l.78-80)

- The mention of the WHO PHSM recommendations and the survey tool implementation in Bosnia appears twice. It could be streamlined to avoid repetition.

>> We shortened the last paragraph of the introduction, including the information about PHSM and BI studies in the paragraph above:

„ The aims of the survey were to 1) monitor the behavioral determinants critical for population compliance with the PHSM, including health literacy, risk perceptions, trust, and perception of fairness of the measures; 2) assess population behavioral changes regarding uptake of PHSM over time; 3) see how those changes relate to behavioral determinants, and contribute to findings on factors encouraging PHSM uptake more broadly; and 4) identify socio-demographic factors (such as sex and age) that help to describe target groups with low compliance and low trust.

This study scrutinizes the results of the behavioral monitoring system in conjunction with self-reported behaviors. The paper begins by outlining the research methodology including the survey measures. Section II analyses the survey data across the five waves. Section III discusses the key behavioral determinants of PHSM uptake in FBIH. Section IV concludes and offers recommendations.“ (p. 4, l. 84-93)

Methods:

The sentence starting with "Time of data collection" (lines 96-98) is a bit awkward and can be restructured for better readability. It might flow better if the dates for each wave are listed in a tabular format or set apart as bullet points.

>> We included the reviewers suggestion and changed the sentence into a list of bullet points, clearly improving the paragraphs readability:

“ Sample and procedure

The research team fielded five cross-sectional survey waves in the FBIH during different periods of the COVID-19 pandemic in the country between June 2020 and August 2021, reaching a total of N=5,195 participants, approximately 1,000 unique respondents per wave (12).

Time of data collection:

-first wave 05/06/2020 - 08/06/2020;

-second wave 03/07/2020 - 06/07/2020;

-third wave 19/09/2020 - 22/09/2020;

-fourth wave 04/12/2020 - 07/12/2020; and

-fifth wave 05/08/2021 - 08/08/2021. “ (p.4, l. 99-104)

-The response rate section (lines 102-105) is informative but could be made clearer with some rewording. The phrasing feels slightly redundant and could use more precision in presenting the varying rates.

>> We rephrased the sentences, and now state more precisely:

„The research company collecting the data distributed the online questionnaires to various population segments, enabling collection of a diverse range of respondents over time. While distribution methods were constant, response rates increased in the later waves (Wave 1: 21%, 4.818 invitations; Wave 4: 54%, 1.958 invitations; Wave 5: 47%, 2.158 invitations).” (p. 5, l. 117-119)

- The sentence "Dropout rate for all five waves were 8%" (line 109) should have more context on what this means for the study. Does this reflect a good retention rate? Additionally, "were" should be corrected to "was."

>> We clarified that we included participants in the dropout count, that ended the questionnaire before giving all answers now in the manuscript:

“Participants were allowed to end the questionnaire at any time. However, over all five waves only 8% of participants who started the questionnaires dropped out before giving all answers.” (p.5, l.123-124)

- It would help to clarify why there was such a variation in the number of people invited to each wave and the response rates. It might indicate something about the survey methodology or external factors.

>> As we reported these invitation numbers, we decided to discuss possible consequences for the data interpretation (and further recommendations for future studies) in the limitations section of the data:

As a further limitation, we experienced a difference in the invitation / participation ratio over the waves, with more people needed to be invited at the earlier waves of the cross-sectional survey. This might be caused by the intense information density at the earlier stages of the pandemic. In consequence, people may have been striving more for distraction, producing a self-selection bias in the early waves of our data. The people who were affected most strongly by the pandemic may have evaded the questionnaire study. Future pandemic studies with special panels could carefully address precisely these people, e.g. with help hotlines or counseling services, including qualitative approaches alongside the quantitative data for the general population. With this approach, behavioral insights could also be used for this target group to adapt pandemic decisions and support even better to the people who are psychologically affected most. (p.27, l. 534-543)

- Remove unnecessary details: For example, in the "Health literacy" section, you mention internal consistency in both waves, but this could be summarized without repeating the Cronbach's alpha scores twice.

>> We shortened the respective paragraph to:

“Health literacy. In waves 2-4, participants were asked about their perceived ability to find the information they needed about symptoms, treatments, and behavioral recommendations for COVID-19. In wave 5, this scale was extended with six items to COVID-19 vaccination literacy. Respondents rated their agreement on a 7-point scale ranging from 1 (strongly disagree) to 7 (strongly agree). Cronbach’s alpha indicates excellent internal consistency (α =.88 [.88;.89]) in all waves. (p.6-7; l. 154-162)

>>Also, we deleted the sentences about predictors and determinants from every paragraph in the methods section, because we include the description paragraph about the statistical analyses, avoiding repetition.

- Avoid over-explaining Likert scales: In multiple instances, the explanation of Likert scales (1 to 7) could be simplified. After the first explanation, the reader should understand how the scale works, so reiterating it for every variable is unnecessary.

>> We agree with the reviewer that explaining the scales is bringing a lot of repetition In the methods section of the manuscript. We can simplify the sentences by deleting the “7-point” information. However, we would strongly argument for including the description of the endpoints of the scales in the text for two reasons

(1) They change over the different predictors, having for example 1 spreading slow vs. 7 spreading fast; and 1 (strongly disagree) to 7 (strongly agree)

(2) They indicate what higher (or lower) mean values for participants indicate in terms of content, and they help interpret the direction of the effects (or example higher values mean higher trust and higher emotional response), being precise and replicable.

Given these two implications, we hope that the reviewer can agree with us in this compromise.

- Separate statistical analysis section more clearly: While you do have a statistical analysis section (line 161), the explanations of statistical tests used, like ANOVA and multiple regression, are a bit disjointed. It might help to structure this as a formal, separate subsection that outlines the methodology in a clearer step-by-step manner.

>> We now included a more separated structure for the different analyses. We start in each paragraph with the goal that this analysis procedure has and hope, this is now more understandable. The complete reworked paragraph can be seen under the following comment.

- Explain the statistical tests more clearly: While you describe using one-way ANOVA and Tukey post-hoc tests, it could be helpful to include brief reasoning on why these tests were chosen (i.e., for comparing means across multiple groups) and how they were applied in the context of the study.

>> we included brief descriptions for the statistical tests we used and also made the respective changes from the comment above:

Statistical analysis

First, descriptive statistics were calculated for all demographic characteristics and outcome variables to provide an overview of the sample.

Second, we aimed to examine whether the average values of behavioral determinants changed between the five survey waves. To do this, we performed six one-way analyses of variance (ANOVA), one for each behavioral determinant (self-efficacy, risk perceptions, emotional response, trust, health literacy, and fairness and policy acceptance). ANOVA is a statistical method used to compare means across multiple groups—in this case, the five survey waves—and determine if differences are statistically significant. Tukey post hoc tests were applied to identify specific pairwise differences between the cross-sectional samples. Additionally, we performed an ANOVA for protective behaviors to analyze whether they varied between the five waves.

Third, multiple linear regressions were used to identify the relationship between one dependent variable (e.g., preventive behaviors) and multiple independent variables (e.g., behavioral determinants). To further explore whether socio-demographic characteristics (predictors like age, gender and education) influenced behavioral determinants (depen

---

## [Editor Report · Decision Letter 1]

19 Feb 2025

Behavioral insights during the COVID-19 pandemic in the Federation of Bosnia and Herzegovina: the role of trust, health literacy, risk and fairness perceptions in compliance with public health and social measures

PONE-D-24-28827R1

Dear Dr. Eitze,

We’re pleased to inform you that your manuscript has been judged scientifically suitable for publication and will be formally accepted for publication once it meets all outstanding technical requirements.

Kind regards,

Iskra Alexandra Nola

Academic Editor

PLOS ONE

Additional Editor Comments (optional):

Dear dr Eitze,

Thank you for revised paper. I would like to ask you one more thing - please change the cover letter, you uploaded in the system with revised version of your paper, with the correct Journal name in first paragraph.

Kind regards,

Iskra A. Nola
---

## [Editor Report · Acceptance letter]

PONE-D-24-28827R1

PLOS ONE

Dear Dr. Eitze,

I'm pleased to inform you that your manuscript has been deemed suitable for publication in PLOS ONE. Congratulations! Your manuscript is now being handed over to our production team.

Kind regards,

on behalf of

Dr. Iskra Alexandra Nola

Academic Editor

PLOS ONE